# Cytotoxic Activity of Melatonin Alone and in Combination with Doxorubicin and/or Dexamethasone on Diffuse Large B-Cell Lymphoma Cells in In Vitro Conditions

**DOI:** 10.3390/jpm13091314

**Published:** 2023-08-27

**Authors:** Sylwia Mańka, Piotr Smolewski, Barbara Cebula-Obrzut, Agata Majchrzak, Klaudia Szmejda, Magdalena Witkowska

**Affiliations:** 1Department of Experimental Hematology, Medical University of Lodz, 93-510 Lodz, Poland; syla.manka@gmail.com (S.M.); piotr_smolewski@wp.pl (P.S.); barbara_cebula@wp.pl (B.C.-O.); klaudia.szmejda@o2.pl (K.S.); 2Department of Hematology, Copernicus Memorial Hospital, 93-510 Lodz, Poland; majchrzak_agata@o2.pl

**Keywords:** melatonin, diffuse large B-cell lymphoma, apoptosis, cell cycle, doxorubicin, dexamethasone

## Abstract

Melatonin (MLT), a pineal gland hormone, not only regulates circadian and seasonal rhythms, but also plays an important role in many aspects of human physiology and pathophysiology. MLT is of great interest as a natural substance with anti-cancer activities. The aim of this study was to assess the cytotoxicity and apoptosis of MLT, used alone or in combination with one of the most active anti-cancer drugs, doxorubicin (DOX), and a well-known anti-inflammatory drug, dexamethasone (DEX), on a diffuse large B-cell lymphoma (DLBCL)-derived cell line. The cytotoxicity and cell cycle distribution were measured using propidium iodide staining, while apoptosis was assessed using the annexin-V binding method. Additionally, to elucidate the mechanisms of action, caspase-3, -8, and -9 and a decline in the mitochondrial potential were determined using flow cytometry. MLT inhibited cell viability as well as induced apoptosis and cell cycle arrest at the G0/G1 phase. The pro-apoptotic effect was exerted through both the mitochondrial and caspase-dependent pathways. Furthermore, we observed increased cytotoxic and pro-apoptotic activity as well as the modulation of the cell cycle after the combination of MLT with DOX, DEX, or a combination of DOX + DEX, compared with both drugs or MLT used alone. Our findings confirm that MLT is a promising in vitro anti-tumour agent that requires further evaluation when used with other drugs active against DLBCL.

## 1. Introduction

Worldwide, diffuse large B-cell lymphoma (DLBCL) is the most common type of non-Hodgkin lymphoma (NHL), accounting for between 30% and 40% of newly diagnosed cases [1]. At the time of diagnosis, the median age is 65 years and the risk of developing DLBCL increases with age [2]. On the basis of the gene expression profile, two major molecular subtypes of DLBCL, germinal centre B-like (GCB) and activated B-like (ABC), have been defined. Subtypes of DLBCL have different clinical outcomes and expected treatment responses, i.e., the overall survival of patients with the ABC gene expression profile is significantly worse than those with GCB [3]. The standard first-line treatment for DLBCL is based on a multidrug chemotherapy protocol with the R-CHOP regimen (rituximab, cyclophosphamide, hydroxydaunorubicin hydrochloride—doxorubicin, Oncovin^®^—vincristine, and prednisone). The standard therapy is effective in approximately 60% of cases [1].

Melatonin (MLT) is a hormone synthesized mainly by the pineal gland as well as by immune cells (lymphocytes, bone marrow, and platelets), the retina, the skin, and the gastrointestinal tract, affecting these cells through paracrine signalling [4,5]. Besides coordinating circadian and seasonal rhythms, which is the main function of MLT, it plays diverse and multiple roles in human physiology and pathophysiology. Nowadays, MLT is not only a hormone, but also an endogenous substance with anti-oxidant, anti-tumour, and immunomodulatory properties [4]. Hence, a strong trend has been observed towards the therapeutic possibilities of this hormone and its potential application in the treatment of many diseases and clinical conditions, such as cardiovascular, gastrointestinal, infectious, psychiatric, neurological, and neurodegenerative conditions as well as cancers [6,7].

Of note, disturbances in MLT synthesis and secretion contribute to the development and/or progression of certain disorders, including cancers. One of the reasons for an increased incidence of neoplastic diseases in elderly people can be the declined immunity associated with the age-related impaired production of MLT [8,9]. In addition, in a wide variety of diseases, such as neurodegenerative diseases (e.g., Alzheimer’s disease), neurological and psychological conditions (e.g., autism, schizophrenia, migraines, and bulimia), metabolic conditions (e.g., type 2 diabetes), cardiovascular disorders (e.g., myocardial infarctions), and, in some cases, cancer, reduced MLT levels are observed [10]. As for haematological malignancies, Rana et al. reported that patients with chronic lymphocytic leukaemia (CLL) exhibit significantly reduced levels of MLT apart from changes in the expression of some circadian “clock genes” compared to healthy subjects, which suggests that the decreased production of MLT may elevate the risk of CLL [11]. MLT is secreted by the pineal gland principally during the night, and for this reason, exposure to artificial light at night (ALAN) causes certain disturbances in MLT synthesis and suppresses its secretion [12]. Zhong et al. showed that ALAN has been potentially linked to an increased risk of NHL, particularly DLBCL [13]. Several epidemiological studies have demonstrated an association between shift work (night work) and the development of leukaemia [11,14] or lymphoma [15].

On the other hand, numerous experimental studies have documented the anti-tumour properties of MLT in various types of cancer such as breast, prostate, gastric, colorectal, and liver cancer as well as in haematological neoplasms [16,17]. MLT exerts its widely described anti-cancer properties through the modulation of different biological processes, including apoptosis, the cell cycle, oxidative stress, autophagy, angiogenesis, and immune system actions [18]. Additionally, there are promising reports that combining MLT with conventional anti-cancer drugs (such as cisplatin, vincristine, doxorubicin, or epirubicin) can improve their antineoplastic effects in different cell lines [19,20]. Moreover, the combination of MLT and arsenic trioxide synergistically killed cancer cells through the activation of the apoptotic pathway in human breast cancer cells [21]. Similarly, the combination of MLT and puromycin synergistically increased cell death in breast cancer cells via a reduction in 45S pre-rRNA and the downregulation of the upstream binding factors XPO1 and IPO7, procaspase 3, and Bcl-xL [22].

Hence, one of the goals of our study was to examine the involvement of MLT in the cell viability and apoptosis of the DLBCL-derived Toledo cell line and the possible mechanisms of this action. The second purpose was to evaluate the in vitro effect of the co-treatment of MLT with an R-CHOP regimen component, i.e., doxorubicin (DOX), and dexamethasone (DEX), which is a well-known anti-inflammatory drug. This may allow us to find out whether their interaction can result in the exertion of cytotoxicity in a DLBCL-derived cell line.

## 2. Materials and Methods

### 2.1. Cell Line

Human GCB-like DLBCL, i.e., the Toledo cell line (CRL-2631), was obtained from the American Type Culture Collection (ATCC, Manassas, VA, USA). Prior to the treatment and culture, the cell viability and quantity were determined with trypan blue using a Vi-cell XR Cell Viability Analyzer (Beckman Coulter, Brea, CA, USA). The cell viability before culturing was not less than 95%. Toledo cells were cultured in an RPMI 1640 medium supplemented with 10% heat-inactivated foetal calf serum (FCS) (PAN-Biotech, Aidenbach, Germany), 2 mM L-glutamine, 50 IU/mL of penicillin, and 50 µg/mL of streptomycin (Sigma Aldrich, St. Louis, MO, USA). The cell line was incubated in a humidified atmosphere with 5% CO_2_ at 37 °C.

### 2.2. MLT and Drug Preparation

MLT (Sigma Aldrich) was dissolved in absolute ethanol (EtOH; POCH, Gliwice, Poland) and was freshly prepared for each experiment as a 100 mM stock solution. It was then added into the RPMI 1640 medium to obtain the required concentrations of MLT. The final concentration of EtOH in the culture medium was not greater than 0.5%.

A Doxorubicin-Ebewe injection (DOX, Ebewe Pharma Ges.m.b.H., Unterach, Austria) was used at the final concentration of 10 ng/mL.

A dexamethasone phosphate injection (Dexaven^®^, DEX, 4 mg/mL) was purchased from PharmaSwiss (Prague, the Czech Republic). DEX was added directly to the cells and reached a final concentration of 4 µg/mL in the culture.

### 2.3. Treatment Protocol

In a pilot study, cell cultures were incubated with MLT at concentrations from 0.5 to 2 mM for 24, 48, or 72 h. For further studies, a 48 h time point and the lowest doses of MLT that induced significant cytotoxicity (1 mM and 1.25 mM) compared with the controls were selected. The optimal doses of DOX and DEX used in the experiments were chosen in a similar way.

Cell cultures (0.5 × 10^6^/mL) were incubated under sterile conditions in 75 mL dishes (Nunc, Roskilde, Denmark). The following culture sets, containing cells treated with the study drugs, were prepared: MLT (1 mM and 1.25 mM), DOX (10 ng/mL), DEX (4 µg/mL), and the combination of DOX + DEX alone. Moreover, combinations of MLT (in both concentrations) with DOX, DEX, or DOX + DEX were tested. In addition, culture-medium-only (untreated) and EtOH (solvent) controls were included in each experiment. The solvent control was treated with absolute EtOH diluted with the medium to obtain a final concentration of 0.5% (*v*/*v*) ethanol.

### 2.4. Cytotoxicity Assay

The cytotoxicity was estimated from cell viability by using propidium iodide (PI) staining. In order to achieve this, MLT as well as all drug-treated cultures were washed twice in cold 1% Dulbecco’s phosphate-buffered saline (DPBS, PAN-Biotech) and stained with 10 µg/mL of PI (Sigma Aldrich) for 15 min, at room temperature, in the dark. Cell fluorescence was measured using flow cytometry with the red FL-3 fluorescent filter. The percentage of PI-positive cells (dead cells) in each sample was determined as the cytotoxic index (CI).

### 2.5. Assessment of Apoptosis

*Apoptosis assay:* Apoptosis was evaluated by measuring phosphatidylserine externalization using an Annexin V Apoptosis Detection Kit (Becton Dickinson, San Jose, CA, USA) according to the manufacturer’s specifications. After incubation, the cells were washed twice with cold PBS and suspended in 85 μL of a binding buffer containing 5 μL of annexin-V-FITC and 10 μL of 10 μg/mL PI. The samples were incubated for 15 min, at room temperature, in the dark. After staining, the cells were immediately analysed with a flow cytometer. The percentage of annexin-V-positive cells in each sample was defined as the apoptotic index (AI).

*Analysis of cell cycle:* The cell cycle distribution was determined based on a DNA content measurement of nuclei stained with PI. The cells were fixed using ice-cold 70% EtOH and stored at −20 °C overnight. After centrifuging, they were incubated in a 300 µL solution containing 50 µg/mL of PI and 50 µg/mL of DNase-free RNase A (Sigma Aldrich), for 1 h in the dark, at 4 °C. The relative proportions of cells in each phase of the cell cycle (subG1, G0/G1, S, and G2/M) were evaluated using flow cytometry and calculated as a percentage of all the cells in the cycle.

*Expression of caspase-3, -8, and -9:* To detect caspase-3 activation, FITC-conjugated rabbit monoclonal antibodies against the anti-active form of caspase-3 were used (BD Pharmingen, San Diego, CA, USA). After incubation, the cells were fixed and permeabilized using a Cytofix/CytopermTM solution (BD Pharmingen) for 20 min on ice. Then, they were washed twice and re-suspended in Perm/WashTM buffer (BD Pharmingen) with anti-caspase-3 antibodies. After a 30 min incubation at room temperature, the cells were washed in Perm/WashTM buffer and immediately analysed using flow cytometry with a fluorescent filter—FL1. In order to detect caspase-8 and -9 activity, FAM-LETD-FMK FLICA^®^ Caspase 8 and FAM-LEHD-FMK FLICA^®^ Caspase 9 Assay Kits were used (Immunochemistry Technologies, Davis, CA, USA). The analysis was performed according to the producer’s protocol and was evaluated using flow cytometry.

*Assessment of mitochondrial membrane potential (ΔΨ):* The loss of the mitochondrial membrane potential was detected using MitoTracker ™ Red CMXRos (MolecularProbes, Eugene, OR, USA). The stock solution of the reagent (1 mM) was diluted in the medium to obtain a working concentration of 50 nM, and 2.5 mL was added to the cell cultures. After a 15 min incubation (humidified atmosphere with 5% CO_2_ at 37 °C), the drop in the ΔΨ was detected using flow cytometry with a fluorescent filter—FL3.

### 2.6. Flow Cytometry Analysis

All fluorescence measurements were performed using flow cytometry (FACSCanto, Becton Dickinson, San Jose, CA, USA) using standard emission filters: green for FITC (FL1, λ = 530 ± 20 nm), orange for PE (FL2, λ = 560–600 nm), and red for Cy-5 (FL3, λ > 600 nm), where necessary. Ten thousand cells per sample were acquired for each analysis.

### 2.7. Statistical Analysis

All the obtained results were analysed with a statistical software (Statistica 13.3, Tibco Software Inc., USA). The data are presented as the mean ± standard error of the mean (SEM) of at least five independent experiments. The normality of the data distribution was checked using the Shapiro–Wilk test. Significant differences between the experimental groups were compared using a one-way ANOVA test followed by a post hoc Tukey test, and *p* < 0.05 was adopted as the threshold for statistical significance.

## 3. Results

### 3.1. Cytotoxic Effect of MLT Alone and in Combination with Drugs on Toledo Cells

As shown in Figure 1, MLT decreased the cell viability in a concentration- and time-dependent manner. The half-maximal inhibitory concentrations (IC50) of MLT were 3.36 mM, 1.7 mM, and 1.46 mM at 24, 48, and 72 h, respectively. For further experiments, a 48 h incubation time and MLT concentrations of 1 mM (CI = 19.5 ± 0.8%, *p* < 0.001) and 1.25 mM (CI = 28.8 ± 1.4%, *p* < 0.001) were selected as the minimal doses required to induce a significant decrease in cell viability compared with solvent controls.

The treatments with DOX (*p* = 0.001), DEX (*p* = 0.03), or DOX + DEX (*p* < 0.001) significantly reduced the DLBCL-derived cell viability in comparison to untreated controls (Figure 2). There were no significant differences in the viability of cells between the untreated and solvent controls (*p* > 0.05). For MLT applied at a concentration of 1 mM, the combinations of MLT + DOX and MLT + DEX did not markedly increase the cytotoxicity compared to the single drugs. In turn, the combination of MLT + DOX + DEX showed a statistically significantly higher CI value (30.4 ± 1.0%) than MLT used alone (18.7 ± 1.9%, *p* = 0.02). For MLT administered at a concentration of 1.25 mM, the combinations of MLT + DOX, MLT + DEX, and MLT + DOX + DEX exerted a significantly higher cytotoxicity in comparison with each single drug (*p* < 0.01, *p* < 0.001, and *p* < 0.001, respectively). Importantly, the effects of the above-mentioned combinations, i.e., MLT + DOX, MLT + DEX, and MLT + DOX + DEX, were also significantly higher compared to those of MLT alone (*p* < 0.01, *p* < 0.001, and *p* < 0.001, respectively) (Figure 2).

### 3.2. Pro-Apoptotic Effects of MLT Alone and in Combination with Other Drugs

The Ann-V/PI assay demonstrated pro-apoptotic activity for both MLT concentrations in the Toledo cells (Figure 3). The mean AIs were 17.5 ± 1.4% for 1 mM MLT (*p* < 0.01 vs. controls) and 18.3 ± 1.7% for 1.25 mM (*p* < 0.01 vs. controls). Importantly, no statistical differences were observed in the percentage of annexin-V-positive cells between the untreated and solvent controls (*p* > 0.05).

DOX, DEX, and the combination of DOX + DEX induced a significant increase in apoptosis: 19.4 ± 2.2%, 15.0 ± 1.6%, and 24.1 ± 2.8% in comparison to the untreated controls (4.9 ± 0.5%, with *p* = 0.0004, *p* = 0.03, and *p* = 0.0001, respectively). MLT applied at a concentration of 1 mM did not trigger significant apoptosis (*p* > 0.05) in the combinations of MLT + DOX and MLT + DEX. In turn, the pro-apoptotic effect induced by the combination of MLT + DOX + DEX (27.7 ± 2.4%) was higher only in comparison to MLT used alone (*p* = 0.03). Importantly, for MLT administered at a concentration of 1.25 mM, the pro-apoptotic effect induced by the combinations of MLT + DOX, MLT + DEX, and MLT + DOX + DEX was markedly higher than that for each drug and for MLT used alone. MLT + DOX, MLT + DEX, and MLT + DOX + DEX induced the following mean AIs: 31.2 ± 2.1%, 33.1 ± 1.1%, and 38.8 ± 1.8%, respectively (*p* = 0.006 vs. DOX, *p* = 0.0001 vs. DEX, *p* = 0.0003 vs. DOX + DEX, and *p* < 0.01 vs. MLT) (Figure 3).

### 3.3. The Mechanisms of the Pro-Apoptotic Activity of MLT Alone and in Combination with Other Drugs

The mechanism of MLT’s pro-apoptotic action on DLBCL-derived cells was caspase-dependent, triggered through extrinsic as well as intrinsic pathways, because MLT (at both used concentrations) significantly enhanced caspase (-3, -8, and -9) activation (Figure 4A) and decreased the ΔΨ compared with the solvent controls (Figure 4B).

DOX and the combination of DOX + DEX significantly activated caspases (-3, -8, and -9) and increased the percentage of cells with a decline in the ΔΨ of the Toledo cells. In turn, DEX did not change the activation of caspase-8 and -9 (*p* > 0.05), although the expression of caspase-3 and the decline in ΔΨ were significantly higher than in the untreated cells (*p* = 0.04 and *p* = 0.009 vs. control, respectively). For MLT administered at a dose of 1 mM, the combinations of MLT + DOX and MLT + DEX did not change the caspase (-3, -8, and -9) activation or result in a drop in ΔΨ (*p* > 0.05), which is consistent with the results of the apoptosis assay. In the samples treated with the MLT + DOX + DEX combination, elevated caspase-3 and -9 activation and a decline in ΔΨ only in comparison to MLT used alone was observed (*p* = 0.005, *p* = 0.001, and *p* = 0.01, respectively). In turn, for MLT applied at a concentration of 1.25 mM, in the MLT + DOX combination, the percentage of cells with active caspase-3 and -9 was higher than that in the cells treated with DOX used alone (*p* = 0.006 and *p* = 0.04, respectively), whereas the activation of caspase-8 was elevated in comparison to MLT used alone (*p* < 0.05). In addition, a reduction in ΔΨ was observed in comparison to either drug used alone. Meanwhile, for MLT administered at a concentration of 1.25 mM, in the combinations of MLT + DEX and MLT + DOX + DEX, elevated caspase-3, -8, and -9 activation and a decline in ΔΨ in comparison to each drug used alone was demonstrated (*p* < 0.01 for all parameters) (Figure 4).

### 3.4. Effect of MLT Alone and in Combination with Other Drugs on Cell Cycle Distribution of DLBCL-Derived Cells

To examine the mechanism responsible for MLT- and drug-mediated cytotoxicity in Toledo cells, the effect of MLT and the drugs on the cell cycle parameters was determined using a PI assay (Figure 5). MLT highly significantly (*p* < 0.001) increased the percentage of cells in the G0/G1 phase with a mean percentage of 83.6 ± 1.2% (1 mM) and 84.3 ± 2.0% (1.25 mM) in comparison to the solvent controls (67.1 ± 1.3%). Additionally, MLT significantly increased the percentage of the subG0-phase cell population, which is consistent with apoptosis at 1 mM (5.9 ± 0.5%, *p* < 0.001) and 1.25 mM (5.8 ± 0.6%, *p* < 0.002).

DOX and the combination of DOX + DEX resulted in an increase in the cells in the G2/M phase with a corresponding decreased proportion of cells in the G0/G1 phase (*p* < 0.001 vs. untreated control). Importantly, the percentages of subG0-phase cells also increased in comparison to the control group (*p* < 0.001). In turn, DEX led to the accumulation of cells in the G0/G1 phase and a related reduction in the proportion of cells in the S phase in comparison to the untreated controls (*p* = 0.0002 and *p* = 0.0004, respectively). The combination of DOX + DEX highly significantly (*p* < 0.001) resulted in a decrease in the cells in the G0/G1 phase with a corresponding increased proportion of cells in the G2/M phase in comparison to DEX used alone. Consequently, because the cytotoxic and pro-apoptotic effects induced by the combination of MLT with drugs were only noticeable and statistically significant for MLT administered at a dose of 1.25 mM, we preferred to choose this concentration to elucidate the effect of the combination treatment on the cell cycle (Figure 5). For MLT administered at a concentration of 1.25 mM, the combinations of MLT + DOX and MLT + DOX + DEX highly significantly (*p* < 0.001) reduced the proportion of cells in the G2/M phase and simultaneously increased the number of cells in the subG0 phase in comparison to either drug used alone. In turn, the combination of MLT + DEX decreased the number of cells in the G0/G1 phase (*p* < 0.001 vs. both drugs), whereas it significantly (*p* < 0.001) increased the proportion of cells in the G2/M phase in comparison to samples treated with DEX or MLT alone. Interestingly, the combination of MLT + DOX + DEX significantly increased the proportion of cells in the G2/M phase in comparison to the MLT + DOX (*p* < 0.001) and MLT + DEX (*p* < 0.02) treatments.

## 4. Discussion

It is well known that MLT plays an important role in a wide range of immunoenhancing actions. Moreover, its oncostatic properties involve a variety of cancer-inhibitory processes. In recent years, due to its promising effects on the prognoses of patients with cancer, MLT has attracted attention, and it is considered a novel drug that can be used alone or in supportive therapy [23].

In cancers, the apoptotic pathway is typically inhibited through numerous means, which makes it a popular target for anti-cancer therapy [24]. The impact of MLT on apoptosis seems to be very attractive for oncological research because it promotes apoptosis in various cancer cells and inhibits cell death in healthy ones. The exact mechanism responsible for this phenomenon is still unclear [25]. Numerous studies have documented that MLT has powerful anti-proliferative and pro-apoptotic properties in various neoplasms, including breast, prostate, gastric, and liver cancers as well as haematological malignancies [16,17]. MLT’s effects on cell proliferation and apoptosis have been extensively studied in vitro in haematological malignancies. However, it has been largely restricted to some cell lines, i.e., HL 60 (myeloid leukaemia), Jurkat (acute T leukaemia), and Ramos (Burkitt’s lymphoma) cells [21,26,27,28]. Yet, the effects and mechanisms of the MLT action on DLBCL-derived cells have been poorly tested and elucidated.

In our study, we firstly noted that MLT at both the tested concentrations (1 mM and 1.25 mM) resulted in a decrease in Toledo cell viability in a dose- and time-dependent manner, as well as the promotion of apoptosis. We showed caspase-3, -8, and -9 activation as well as a distinct decrease in the mitochondrial membrane potential (ΔΨ). The mechanism responsible for MLT-induced cell death appears to depend on both mitochondrial and caspase-activation pathways. Similar results have been demonstrated in studies conducted by other authors in which in vitro apoptosis increased significantly in a variety of haematological neoplasms, such as both acute and chronic myeloid leukaemia, acute lymphoid leukaemia, and lymphoma, after MLT treatment [29,30]. Sánchez-Hidalgo et al. demonstrated that MLT (2 mM) intensified both cell death and cycle arrest, although the tested cell lines exhibited different sensitivities to the hormone. The Ramos (Burkitt’s lymphoma) and DoHH2 (follicular lymphoma) cells were the most sensitive, whereas SU-DHL-4 cells (DLBCL) demonstrated a moderate sensitivity to MLT. Finally, Jurkat cells appeared to be the least sensitive. Unfortunately, the reason for this ambiguous response to the hormone has not been explained [30]. In our centre, we also noted marked variations in the responses to MLT among the tested malignant lymphoid cell lines. The cell sensitivity, starting with the highest, was as follows: RPMI 6666 (Hodgkin lymphoma), Toledo (DLBCL), and RAJI (Burkitt’s lymphoma) cells (data not published).

The authors suggested that MLT demonstrated a moderate cytotoxic effect and/or enhanced apoptosis in lymphoma- and leukaemia-derived cell lines via its pro-oxidant properties [27,31] and via the melatonin membrane receptor independent pathway [26,32]. It was also demonstrated that MLT downregulates the expression of anti-apoptotic bcl-2 as well as increases the expression of pro-apoptotic bax [26,28,30]. Our study showed that both the intrinsic and extrinsic pathways of apoptosis may be activated by MLT in the DLBCL cell line. Similarly to us, Sanchez-Hidalgo et al. observed that an MLT (2 mM) treatment in Burkitt- and DLBCL-derived cell lines contributed to the activation of caspase-8 and, simultaneously, to caspase-3 and -9 activation, as well as a reduction in the ΔΨ. Due to selective caspase-8 activation, which was detected only in the above-mentioned study, but not in the other tested cell lines, the authors suggested that the activation of the intrinsic pathway could be critical for MLT-induced apoptosis in lymphoid malignancies, and the extrinsic pathway does not seem necessarily involved [30].

Uncontrolled cell proliferation is the hallmark of cancer, and tumour cells are directly regulated by the cell cycle [33]. Hence, we revealed that MLT caused the significant G0-/G1-phase arrest of the cell cycle. This indicates that one of the mechanisms through which MLT destroys cancer cell viability is through the inhibition of cell cycle progression. Our data are in concordance with those reported for non-small cell lung cancer, ovarian cancer, gastric cancer, and some haematological cancer cells that were treated with MLT and resulted in similar G0/G1 arrest, the inhibition of cell proliferation, and the induction of apoptosis [28,30,34,35,36]. The accumulation of MLT-treated cells in the G0/G1 phase might be due to the suppression of the expression of cell-cycle-related proteins (CDK4 and CDK2) and the regulation of diverse signalling pathways involved in cell proliferation, differentiation, and apoptosis such as extracellular signal-regulated kinase 1/2 (ERK 1/2) or phosphatidyloinositol 3-kinase (PI3K)/Akt/mammalian target of rapamycin (mTOR) [34,35,37]. In contrast, the data showed that, in hepatocarcinoma, neuroblastoma, and acute T-cell leukaemia cell lines, MLT increased the cell population in the G2/M phase [38,39,40]. These differences in cell cycle distribution may be related to the concentration of the hormone used in the experiments, and may also depend on the types of tested cancer cell lines.

Recently, an increased number of reports have described the beneficial effects of MLT administered in combination with numerous well-known anti-tumour agents such as vincristine [20], DOX [21,41], docetaxel [42], epirubicin [22], and cisplatine [19]. In addition, the synergistic effects of MLT in combination with new agents (recently approved for clinical use or still in clinical trials) such as everolimus (mTOR inhibitor) or barasertib (selective inhibitors of aurora B kinase) on the cytotoxicity and induction of leukaemia lymphocyte apoptosis have been described [21]. In addition, the use of MLT in vivo in unconventional cancer treatments, such as in the Di Bella programme, is also promising, but debatable. This multitherapy is based on the administration of a low dose of cyclophosphamide, which is a combination of melatonin, somatostatin, bromocriptine, retinoids, and the adrenocorticotropic hormone, to improve the drug efficacy on tumour cells and the patients’ quality of life [43]. The mechanism by which MLT sensitises tumour cells to other drugs, thus being responsible for the synergistic or additive effects, is not fully understood. In some publications, the authors suggested that the resistance of the tumour cells to anti-cancer drugs, such as epirubicin or DOX, is overcome by the impact of MLT on the inhibition of P-glycoprotein expression via the nuclear factor kappa B (NF-kB) pathway [22,44,45]. Interestingly, those above-mentioned synergisms or additive effects of MLT occur with a variety of anti-cancer drugs with completely different mechanisms of action.

The effect of MLT in combination with conventional chemotherapeutic drugs seems to be multidirectional. On the one hand, it protects against drug-induced toxicity, but on the other hand, the hormone increases the sensitivity of cells to drugs and enhances their therapeutic effect [46]. Similar beneficial results for MLT were observed in our study. As the R-CHOP regimen is currently a first-line treatment for DLBCL patients, we assessed the interaction of MLT used alone and in combination with two components of R-CHOP, DOX and DEX. We observed that DOX and the combination of DOX + DEX significantly activated caspase-3, -8, and -9 and increased the percentage of cells with a decline in the mitochondrial membrane potential compared to the control. This finding is in line with other studies showing that DOX, similarly to other anti-cancer drugs, induced the caspase- and mitochondrial-dependent apoptosis pathways [47]. Moreover, we noted a significant potentiation of cytotoxicity and apoptosis for DOX, DEX, and the combination of both compounds towards DLBCL-derived cells when combined with MLT in comparison to each drug and MLT applied separately. Although our results might indicate an interesting therapeutic target for DLBCL patients, thus far, there are no similar data to compare with in the published literature. Apoptosis induced by a combinational treatment with MLT + DOX, MLT + DEX, or MLT + DOX + DEX is triggered through a caspase-dependent mechanism via the mitochondrial pathway, as we showed through caspase-3, -8, and -9 activation, and moreover, through a distinct decrease in ΔΨ. Importantly, the effects of MLT obtained with the tested drugs and observed in our study were only noticeable for MLT administered at a dose of 1.25 mM.

In our study, the treatment with DOX and the combination of DOX + DEX in DLBCL cells contributed to the accumulation of cells in the G2/M phase. Interestingly, the combination of MLT + DOX and MLT + DOX + DEX increased the proportion of cells in the subG0 phase and simultaneously shifted cell accumulation from the G2/M to the G0/G1 phases. These facts suggest that MLT rescued the G2/M cell cycle arrest induced by DOX and improved the cytotoxic effect of DOX by changing the inhibition of the G2/M cell cycle phase to the G0/G1 phase. In turn, and similarly to other researchers [48], we showed that a treatment with DEX in DLBCL-derived cells caused G0/G1 phase cell cycle arrest, whereas an MLT + DEX co-treatment increased the proportion of cells in the G2/M phase in comparison to the samples treated with DEX alone.

Our study showed that a combinational treatment with MLT + DOX, MLT + DEX, or MLT + DOX + DEX modulated cell cycle arrest in different ways. It seems that the application of MLT with DOX or DOX + DEX increases the cytotoxic activity of the drugs through apoptosis, but not through G2/M cell cycle arrest. In contrast, the synergistic effect of MLT in a co-treatment with DEX occurs both through apoptosis and G2/M cell cycle arrest.

In conclusion, we want to confirm that MLT, used alone or in combination with drugs routinely used in DLBCL therapy and DEX, has demonstrated toxicity against an investigated cell line through the induction of apoptosis and the modulation of the cell cycle phases. Apart from that, our study provides evidence that MLT is a very attractive substance, both as a monotherapy or as an adjuvant in the treatment of DLBCL. However, more studies evaluating the effects and underlying mechanisms of MLT in combination with other components of the R-CHOP regimen are required.

## Figures and Tables

**Figure 1 jpm-13-01314-f001:**
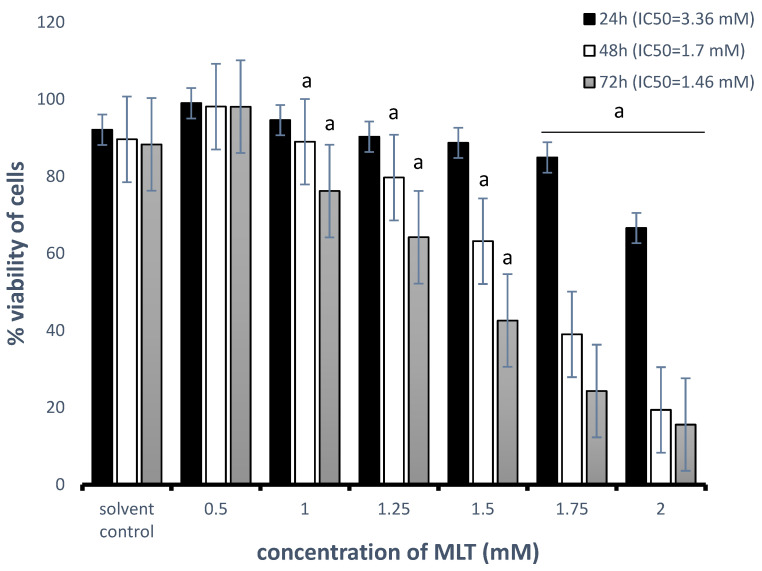
Effects of MLT treatment on the percentage of viable Toledo cells. Cells were treated with 0.5–2 mM MLT for 24 h, 48 h, or 72 h. Cytotoxicity was determined using a PI assay as described in Section 2—“Materials and Methods”. Each bar represents the mean ± SEM, and “a” indicates *p* < 0.05 vs. the solvent control.

**Figure 2 jpm-13-01314-f002:**
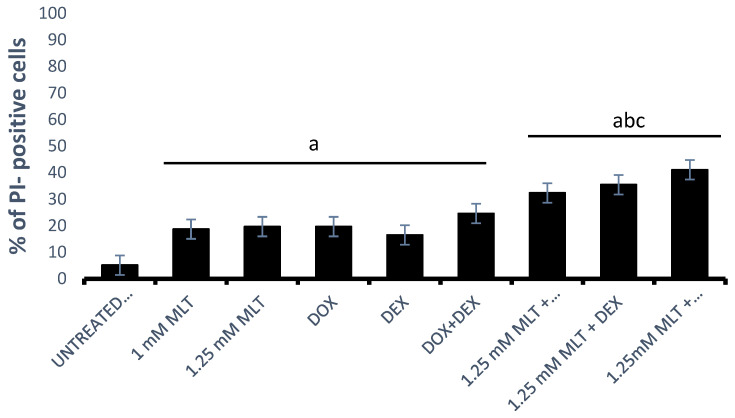
Cytotoxicity of MLT used alone and in combination with DOX, DEX, or a combination of DOX + DEX after 48 h of culture. The cytotoxic effect was determined using a PI assay as described in Section 2—“Materials and Methods”. The data represent the mean ± SEM, “a” indicates *p* < 0.05 vs. the untreated control, “b” indicates *p* < 0.05 vs. the drug used alone, and “c” indicates *p* < 0.05 vs. MLT used alone.

**Figure 3 jpm-13-01314-f003:**
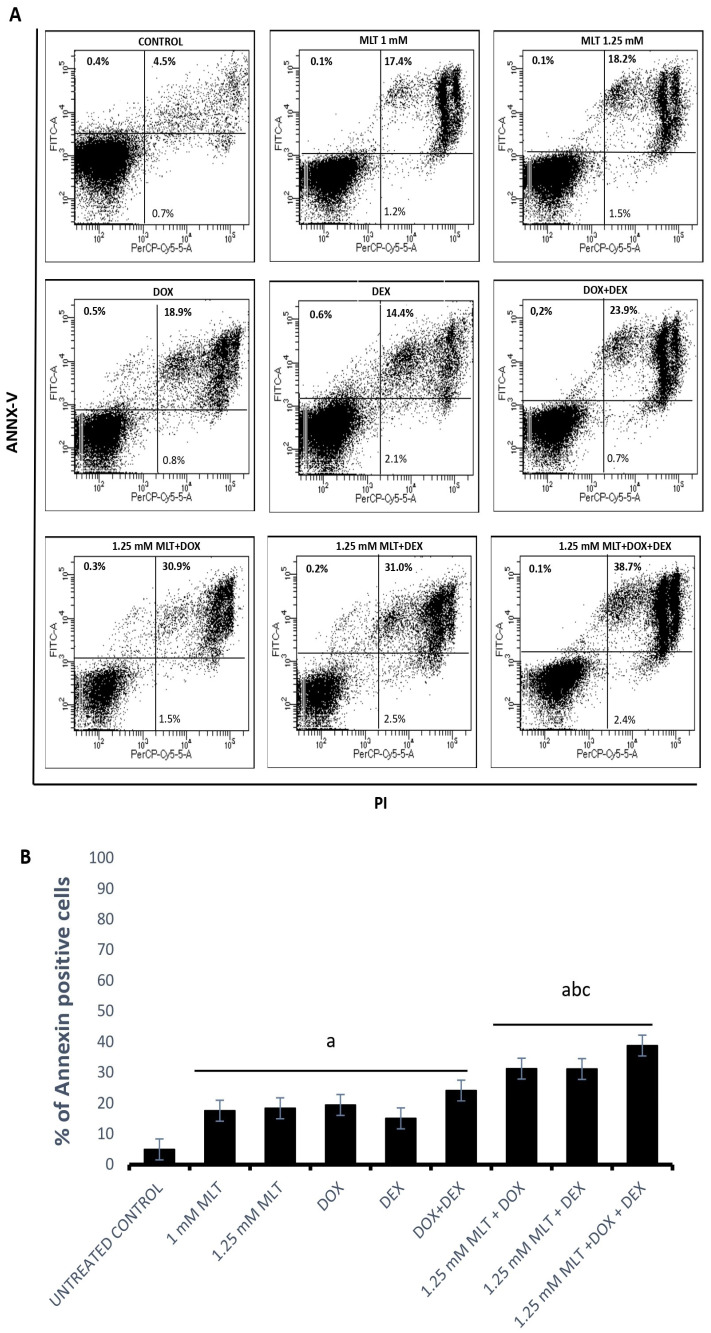
Pro-apoptotic effect of MLT used alone and in combination with DOX, DEX, or a combination of DOX + DEX. Apoptosis (both early and late apoptosis) in the Toledo cell line after 48 h of culture was determined by an Ann-V/PI assay as described in Section 2—“Materials and Methods”. (**A**) Representative results from the flow cytometer analysis. (**B**) Percentage of total apoptotic cells (upper right and left of the panels). Each bar represents the mean ± SEM, “a” indicates *p* < 0.05 vs. the untreated control, “b” indicates *p* < 0.05 vs. the drug used alone, and “c” indicates *p* < 0.05 vs. MLT used alone.

**Figure 4 jpm-13-01314-f004:**
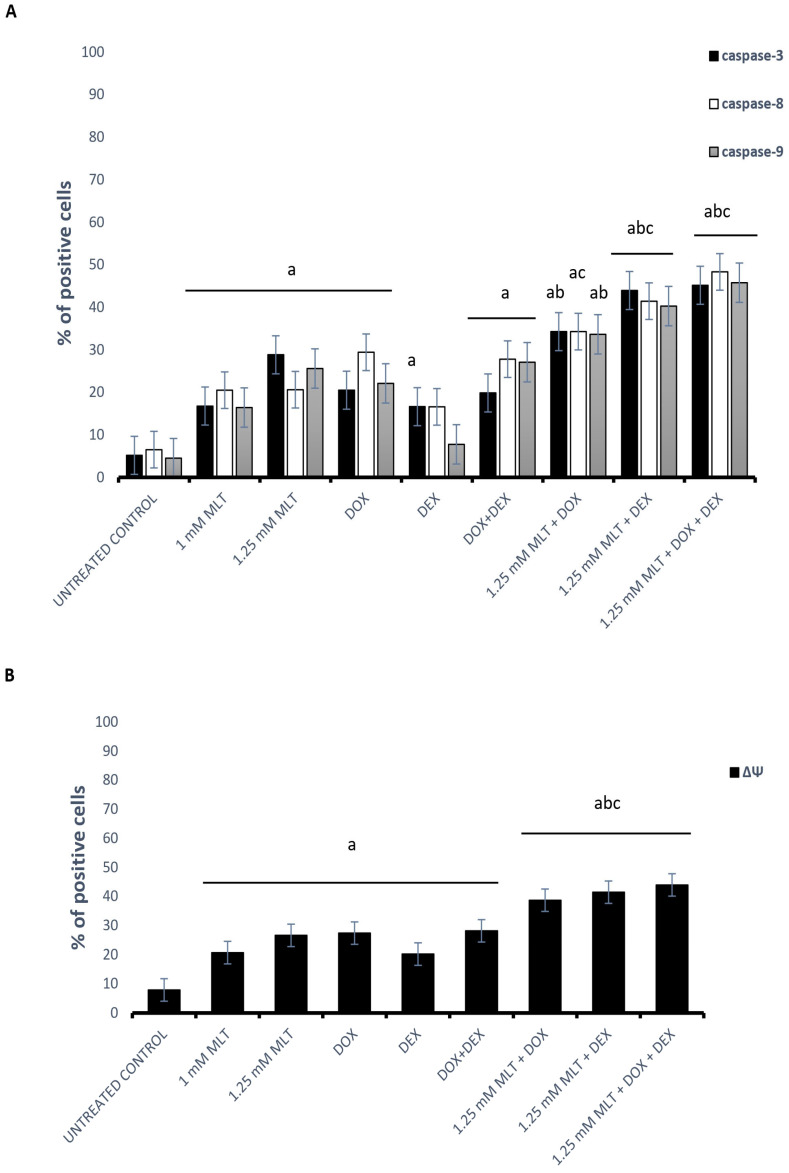
Mechanisms of pro-apoptotic activity of MLT treatment used alone and in combination with DOX, DEX, or a combination of DOX + DEX. Each bar represents the mean ± SEM percentages of positive cells with the activation of caspase-3, -8, and -9 (**A**) and cells with a decline in the mitochondrial membrane potential (ΔΨ) (**B**); “a” indicates *p* < 0.05 vs. the untreated control, “b” indicates *p* < 0.05 vs. the drug used alone, and “c” indicates *p* < 0.05 vs. MLT used alone.

**Figure 5 jpm-13-01314-f005:**
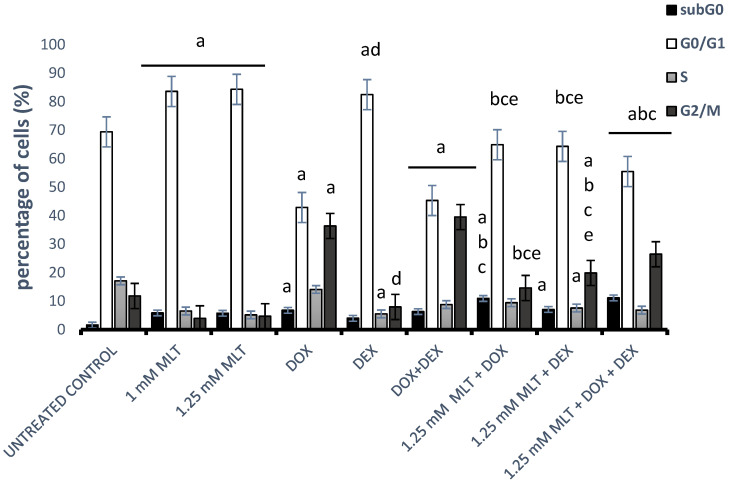
Effect of MLT treatment used alone or in combination with DOX, DEX, or a combination of DOX + DEX on the cell cycle distributions of Toledo cells. The cell cycle analysis was conducted as described in Section 2—“Materials and Methods”. Each bar represents the mean ± SEM, “a” indicates *p* < 0.05 vs. the untreated control, “b” indicates *p* < 0.05 vs. the drug used alone, “c” indicates *p* < 0.05 vs. MLT used alone, “d” indicates *p* < 0.05 vs. the DOX + DEX treatment, and “e” indicates *p* < 0.05 vs. 1.25 mM MLT + DOX + DEX.

## Data Availability

Not applicable.

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
