# Peer review of "Cytotoxic Activity of Melatonin Alone and in Combination with Doxorubicin and/or Dexamethasone on Diffuse Large B-Cell Lymphoma Cells in In Vitro Conditions"

_jpm, 2023, doi:10.3390/jpm13091314_

Round 1
Reviewer 1 Report
The authors need to improve the article to publish as followings,
1. Add control bars for figure 1.
2. Add statistics as followings for all figures and explain
a. DOX vs DOX+DEX
b. DEX vs DOX+DEX
c. 1.25 MLT+DOX vs 1.25 MLT+DOX+DEX
d. 1.25 MLT+DEX vs 1.25 MLT+DOX+DEX
3. Label clearly in all figures: 1 MLT to 1mM MLT, 1.25 MLT to 1.25mM MLT
4. Maybe delete 1mM MLT data. Why 1mM MLT data shown? Just describing on text should be enough.
5. Adding proposed mechanism for MLT combination will be more appreciation for reader.
Author Response
Dear Reviewers,
On behalf of my Co-authors and myself, I would like to thank you for the editorial decision, and the Reviewers for their comprehensive appraisal and constructive comments on our revised manuscript, entitled "Cytotoxic activity of melatonin alone and in combination with doxorubicin and/or dexamethasone on diffuse large B-cell lymphoma cells in in vitro conditions").
We appreciate the possibility to resubmit our paper for publication in your journal. We have performed a careful revision of the manuscript according to your recommendations and provide a point-by-point response to the reviewers’ comments below. We have taken into account all of the above in the revised version of the manuscript. All changes in the text are marked in yellow.
We thank again the Editor and Reviewers for their input and we hope that the improved version of the manuscript is acceptable for publication in the Expert Opinion On Drug Metabolism and Toxicology.
Sincerely yours
Magdalena Witkowska
1. Add control bars for figure 1. - Control bars were added.
2. Add statistics as followings for all figures and explain
a. DOX vs DOX+DEX
b. DEX vs DOX+DEX
c. 1.25 MLT+DOX vs 1.25 MLT+DOX+DEX
d. 1.25 MLT+DEX vs 1.25 MLT+DOX+DEX
It was explained and statistics was added.
3. Label clearly in all figures: 1 MLT to 1mM MLT, 1.25 MLT to 1.25mM MLT
We labeled all figures more clearly.
4. Maybe delete 1mM MLT data. Why 1mM MLT data shown? Just describing on text should be enough.
We corrected as above.
5. Adding proposed mechanism for MLT combination will be more appreciation for reader.
Mechanism of combination was added to the text
Reviewer 2 Report
Sylwia Mańka et al. reviewed the cytotoxicity and apoptosis of Melatonin (MLT), used alone and in combination with doxorubicin (DOX) and dexamethasone (DEX), on diffuse large B-cell lymphoma (DLBCL) cells. They found that MLT inhibited cell viability and induced apoptosis and cell cycle arrest at the G0/G1 phase. Furthermore, MLT, combination with DOX, DEX, and DOX+DEX further increased cytotoxic and pro-apoptotic activity and modulation of the cell cycle. I believe the results are of interest. However, there are several suggestions need to be addressed before publication.
Major revisions:
1. The authors determined the distribution of cell cycle after MLT alone and combination with DOX, DEX, and DOX+DEX exposure, I believe cell cycle-related proteins should be detected by western blotting.
2. In Figure 3, the images of flow cytometer analysis is twelve in Figure 3A, whereas, the histogram is nine in Figure 3B. Please show all of the results in Figure 3B and improve the image resolution in all figures in this manuscript.
Minor comments:
1. In line 37, R-CHOP first appeared, please provide the full name.
2. In line 89, 50 UI/ml of penicillin should be revised by 50 IU/mL of penicillin. ml should be revised by mL in all the manuscript.
3. In line 91, CO2 and 370C should be revised by CO2 and 37oC in all the manuscript.
4. In line 110, 0.5x106/ml should be revised by 0.5×106/mL.
5. There are some grammar errors, moderate editing of English language required.
Moderate editing of English language required
Author Response
Dear Reviewers,
On behalf of my Co-authors and myself, I would like to thank you for the editorial decision, and the Reviewers for their comprehensive appraisal and constructive comments on our revised manuscript, entitled "Cytotoxic activity of melatonin alone and in combination with doxorubicin and/or dexamethasone on diffuse large B-cell lymphoma cells in in vitro conditions").
We appreciate the possibility to resubmit our paper for publication in your journal. We have performed a careful revision of the manuscript according to your recommendations and provide a point-by-point response to the reviewers’ comments below. We have taken into account all of the above in the revised version of the manuscript. All changes in the text are marked in yellow.
We thank again the Editor and Reviewers for their input and we hope that the improved version of the manuscript is acceptable for publication in the Expert Opinion On Drug Metabolism and Toxicology.
Sincerely yours
Magdalena Witkowska
